# Examination of yield, bacteriolytic activity and cold storage of linker deletion mutants based on endolysin S6_ORF93 derived from *Staphylococcus* giant bacteriophage S6

**Sosuke Munetomo[1], Jumpei Uchiyama[2]\*, Iyo Takemura-Uchiyama[2], Thamonwan Wanganuttara[2], Yumiko Yamamoto[2], Toshihiro Tsukui[3], Hideharu Hagiya[4], Shuji Kanamaru[5], Hideyuki Kanda[1], Osamu Matsushita[2]**

1 Department of Public Health, Graduate School of Medicine Dentistry and Pharmaceutical Sciences, Okayama University, Kita-ku, Okayama, Japan, 2 Department of Bacteriology, Graduate School of Medicine Dentistry and Pharmaceutical Sciences, Okayama University, Kita-ku, Okayama, Japan, 3 Nippon Zenyaku Kogyo Co. Ltd., Koriyama, Fukushima, Japan, 4 Department of Infectious Diseases, Okayama University Hospital, Kita-ku, Okayama, Japan, 5 School of Life Science and Technology, Tokyo Institute of Technology, Yokohama Kanagawa, Japan

\* uchiyama@okayama-u.ac.jp

## Abstract

Methicillin-resistant *Staphylococcus* spp. present challenges in clinical and veterinary settings because effective antimicrobial agents are limited. Phage-encoded peptidoglycan-degrading enzyme, endolysin, is expected to be a novel antimicrobial agent. The enzymatic activity has recently been shown to be influenced by the linker between functional domains in the enzyme. S6_ORF93 (ORF93) is one of the endolysins derived from previously isolated *Staphylococcus* giant phage S6. The ORF93 was speculated to have a catalytic and peptidoglycan-binding domain with a long linker. In this study, we examined the influence of linker shortening on the characteristics of ORF93. We produce wild-type ORF93 and the linker deletion mutants using an *Escherichia coli* expression system. These mutants were designated as ORF93-Δ05, ORF93-Δ10, ORF93-Δ15, and ORF93-Δ20, from which 5, 10, 15, and 20 amino acids were removed from the linker, respectively. Except for the ORF93-Δ20, ORF93 and its mutants were expressed as soluble proteins. Moreover, ORF93-Δ15 showed the highest yield and bacteriolytic activity, while the antimicrobial spectrum was homologous. The cold storage experiment showed a slight effect by the linker deletion. According to our results and other studies, linker investigations are crucial in endolysin development.

## Introduction

*Staphylococcus* spp. are a genus of Gram-positive cocci that can be found on skin and mucosa of humans and animals [1,2]. *S. aureus* and *S. epidermidis* can cause a variety of infections, including skin infections like cellulitis, as well as opportunistic infections such as pneumonia,

**Data Availability Statement:** The orf93 gene designed in this study was deposited to the GenBank (accession no. LC797950).

**Funding:** This work was funded by a joint research fund from Nippon Zenyaku Kogyo Co. Ltd. (https://www.zenoaq.com/). J.U. has received this funding. There is no specific grant number. The funders had no role in the study design, data collection and analysis, decision to publish, or preparation of the manuscript.

**Competing interests:** The authors declare that no competing interests exist.

meningitis, endocarditis, and bacteremia in humans [2,3]. *S. pseudintermedius* and *S. schleiferi* cause pyoderma, otitis, postoperative and urinary tract infections in dogs and cats [4,5]. Methicillin-resistant *S. aureus* (MRSA) and *S. pseudintermedius* (MRSP), in particular, are a significant problem because of the limited effectiveness of current antimicrobial agents [5,6]. Additionally, *S. aureus* less susceptible and unsusceptible to vancomycin, such as vancomycin-intermediate *S. aureus* and vancomycin-resistant *S. aureus*, respectively, have already emerged and become threats in the clinical setting [7]. Thus, there is an urgent need to develop new antimicrobial agents with novel mechanisms of action in clinical and veterinary settings.

Endolysin is a peptidoglycan-degrading enzyme produced by bacteriophage (phage) [8]. Endolysin, together with holin, which is encoded on the phage genome, leads to bacterial cell lysis at the last stage of phage infection [9]. Endolysins pass through the pores formed by holins and break down the peptidoglycan [9]. When treating the Gram-positive bacteria, such as *S. aureus*, with endolysins from the outside, the bacteria are lysed by degrading peptidoglycan because those have peptidoglycan on the outside of their cell walls. Endolysin is now expected to be implemented as an alternative, new, antimicrobial modality, because of its following properties: 1) it exhibits rapid bacteriolytic efficacy [10]; 2) it kills specific bacterial genus/species and does not disturb normal microflora [10]; 3) as the peptidoglycan is essential for bacterial survival, the risk of emergence of endolysin-resistant bacteria is not likely to occur [11]; 4) as the mechanism of action is different from that of conventional antimicrobial agents, it can kill the drug-resistant bacteria [12]; and 5) because protein drugs such as cytokines and lysozyme are already on the market, the conventional drug manufacturing platform can be used to produce the pharmaceutical endolysin [13]. However, natural endolysins often have low yield, antimicrobial activity, and storability, and these characteristics need to be improved to develop them as antimicrobial agents [14–17].

Most of the proteins, including endolysin, have several functional domains, which are characterized as structural units with function. In the protein with multiple domains, the domains are connected by short peptides called linkers. Endolysins against Gram-positive bacteria generally have one or two peptidoglycan catalytic domains at the N-terminus and one peptidoglycan-binding domain at the C-terminus, where all domains are connected by linkers [18]. In recent years, it is considered that the linker is important for protein characteristics because it can control the cooperation of neighboring domains [19–24]. For example, the expression of protein without a linker was not successful in *Escherichia coli* [20]. Changing the codons of the linker improved the expression of fusion protein [21]. The linker length is shown to be an important factor for protein characteristics such as stability [22] activity [23] and catalytic efficiency [24]. However, the influence of linker length on the endolysin characteristics has not been examined well, to our knowledge.

Although a single endolysin is typically found in a phage, the giant phages infecting *Staphylococcus* spp. atypically have two endolysins that are aligned tandemly [25,26]. Giant phage S6 can infect a wide range of *Staphylococcus* spp., and S6_ORF93 (ORF93) and S6_ORF94 are predicted as endolysins [27,28]. In this study, we aimed to examine the influences of ORF93 by controlling linker length to the characteristics, such as yield, bacteriolytic activity, and storability, using the *E. coli* protein expression system.

## Results

### Description of ORF93 and its mutants

According to the protein domain analysis by InterPro, ORF93 was predicted to have two domains [29]: the cysteine, histidine-dependent amidohydrolase/peptidase (CHAP) domain was located from the 4th to the 136th amino acid from the N-terminal end; and the Src

homology-3 (SH3) domain was located from the 163rd to the 265th amino acid (Fig 1A). The CHAP domain hydrolyzes the peptide bond within the peptidoglycan, between the terminal glycine of the pentaglycine cross-bridge and D-alanine, and the SH3 domain binds to the peptidoglycan [30]. According to these predictions, we assumed that the linker that connects the two domains was located from the 137th to the 162nd amino acid. The predicted linker consists of 26 amino acids, containing the basic amino acid of lysine at 35% and the neutral amino acids at 65% (S1 Fig).

Moreover, the structure prediction by AlphaFold2 distinctly showed the two domains and a long linker (Fig 1B) [31]. In the result of the predicted local distance difference test (pLDDT) in the structural prediction, the pLDDT scores, which are the confidence value for structural prediction in AlphaFold2, were overall low at the predicted linker region from the 137th to the 162nd amino acid based on the protein domain analysis. This suggested that the predicted linker region is likely to be flexible (S2 Fig) [32]. In addition, the predicted aligned error (PAE) plot, which provides the estimates of the positional error of the amino acids in a predicted structure, showed that the linker region assumed by the protein domain analysis had greater position errors than the other region (S3 Fig). Thus, the ORF93 region from the 137th to the 162nd amino acid (26 amino acids) was assumed to be a linker.

Based on the results of domain prediction from the protein sequence, four linker deletion mutants of ORF93 were designed (Fig 1C). We shortened the linker of ORF93 by 5, 10, 15, and 20 amino acids in the upstream direction from the 157th amino acid from the N-terminus. These four ORF93 mutants were designated as ORF93-Δ05, ORF93-Δ10, ORF93-Δ15, and ORF93-Δ20, respectively. The protein sequences of the ORF93 mutants are shown in S1 Fig. Observing the predicted protein structures of the linker deletion mutants, the superpositions of the predicted structures of ORF93-Δ05, ORF93-Δ10, and ORF93-Δ15 matched each other except for linkers (Fig 1D). The orientation of two domains of ORF93-Δ20 structure was different from those of ORF93 and the other linker deletion mutants (Fig 1E), which corresponds that the deleted region reached the region with higher pLDDT scores than the main linker region with low pLDDT scores (S2 Fig).

## Production of ORF93 and its mutants

The ORF93 and deletion mutants tagged with 6×His N-terminally were overexpressed in *E. coli* and purified with $Co^{2+}$ affinity chromatography. Three out of four mutants, ORF93-Δ05, ORF93-Δ10 and ORF93-Δ15, were expressed in soluble fractions in *E. coli*, while ORF93-Δ20 became insoluble in *E. coli*. The obtained proteins were subjected to the sodium dodecyl sulfate-polyacrylamide gel electrophoresis (SDS-PAGE) analysis. The molecular masses of ORF93, ORF93-Δ05, ORF93-Δ10 and ORF93-Δ15 were estimated to be 28.6 kDa, 27.9 kDa, 27.4 kDa, and 26.9 kDa, respectively (Fig 2A); the theoretical molecular masses were 31.3 kDa, 30.7 kDa, 30.2 kDa, and 29.7 kDa, respectively. Moreover, the purities of obtained proteins were examined using the SDS-PAGE images. The faint minor bands disappeared, as the linker was shortened. The proportions of endolysin contained in the protein solutions were calculated to be 80.2 ± 0.3% at ORF93, 81.1 ± 0.5% at ORF93-Δ05, and 81.5 ± 0.2% at ORF93-Δ10, and 89.8 ± 0.6% at ORF93-Δ15 (mean ± standard deviation (SD); n = 3). Based on these results, ORF93 and the three mutants could be used in further experiments in this study.

Next, we compared the yields of three mutants with that of ORF93. We quadruplicated the experiment in which four protein productions from bacterial culture to protein purification were simultaneously carried out under the same conditions. For the comparison, the yield of wild-type ORF93 was set as 100%, and the relative yields of the other three proteins were calculated. Comparing the yield of ORF93-Δ15 with those of the other three proteins, ORF93-Δ15

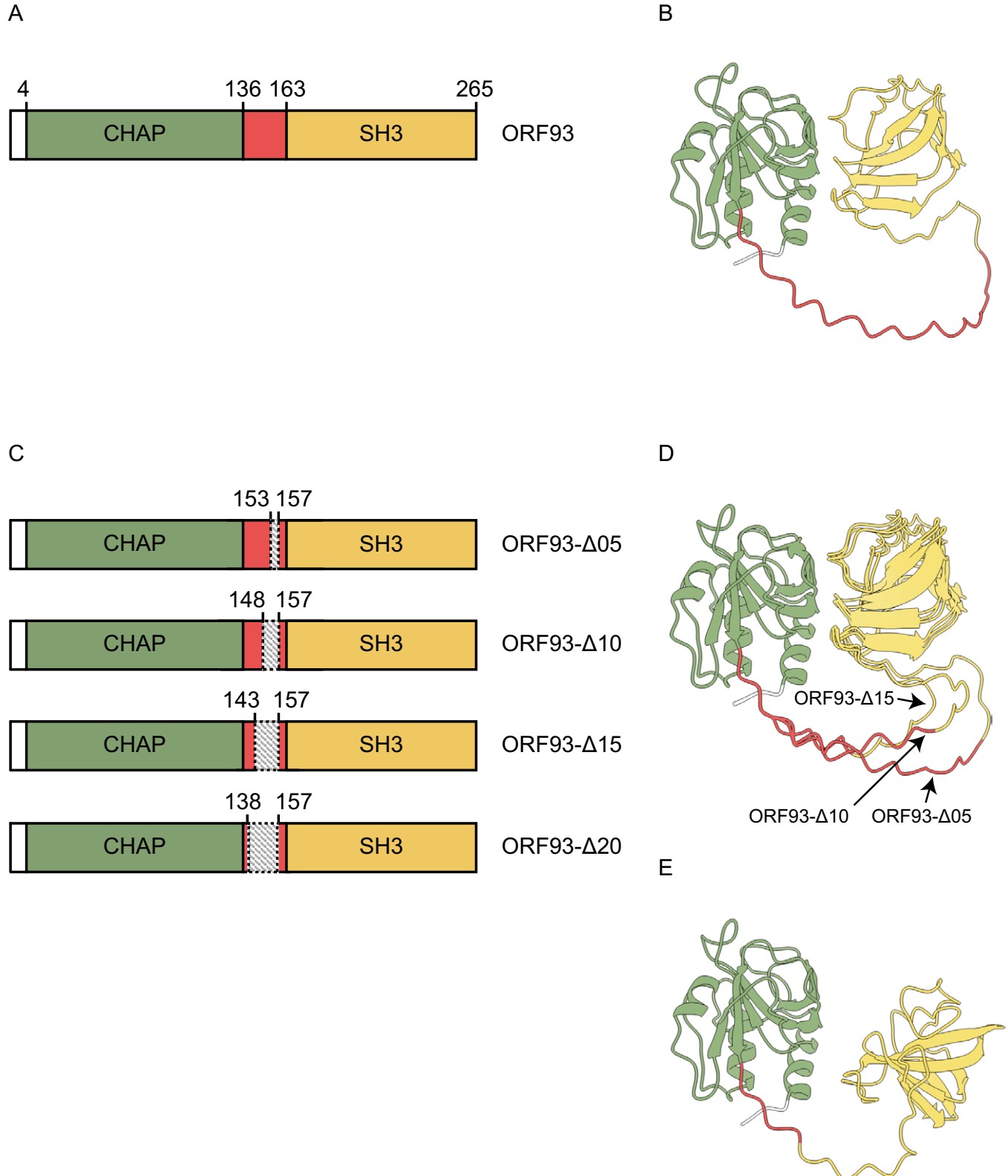

**Fig 1. Description of ORF93 and its mutants.** The domains and linker are colored as follows: CHAP domain in green, linker in red, and SH3 domain in yellow. (A) Schematic diagram of ORF93 based on the primary structure. Each number represents the amino acid position from the N-terminus at both ends of the domain, counted from the N-terminus. (B) Predicted protein structure of ORF93. (C) Schematic diagram of ORF93 mutants based on the primary

structure. The linker deletions are shown as shadowed areas. The numbers around shadowed areas represent the amino acid position from the N-terminus at both ends of the deletion area. ORF93-Δ05, ORF93-Δ10, ORF93-Δ15, and ORF93-Δ20 are the ORF93 mutants which 5, 10, 15, and 20 amino acids were deleted in the upstream direction from the 157[th] amino acid from the N-terminus. The predicted linker lengths are 21, 16, 11, and 6 amino acids, respectively. (D) Predicted protein structures of ORF93-Δ05, ORF93-Δ10 and ORF93-Δ15. These protein structures were overlapped based on the CHAP domain. (E) Predicted protein structure of ORF93-Δ20.

had ca. 2.3, 2.0, and 1.7 times higher yields than ORF93, ORF93-Δ05, and ORF93-Δ10, respectively ($P < 0.05$; Fig 2B). Moreover, we separately triplicated the experiments to calculate the yield in one liter bacterial culture in this culture setting for ORF93 and ORF93-Δ15. ORF93 and ORF93-Δ15 were produced at $1.2 \pm 0.2$ and $2.6 \pm 0.3$ mg/L (n = 3), respectively. Considering these results, the optimal linker length of ORF93 was considered to not only improve the purity but also enhance the yield in *E. coli* overexpression system.

## Antimicrobial spectrum and bacteriolytic activity

The antimicrobial spectra of ORF93 and three deletion mutants were examined by a double-layered agar plate assay. In the assay, the endolysin solutions (3 μL of 0.1 mg/mL) were applied on the surface of double-layered culture media containing bacteria, and the transparency of the spot formed on the agar plate was examined after incubation. In this assay, we tested 11 bacteria, including nine strains of *Staphylococcus* spp., one strain of *Bacillus subtilis*, and one strain of *E. coli*. We also semiquantitatively estimated the strength of antimicrobial activity by observing the degree of transparency of the spot formed on the agar plate at a 4-level scale (i.e., strong, moderate, weak, and no activity; S4 Fig). First, observing the sensitivity to ORF93 and

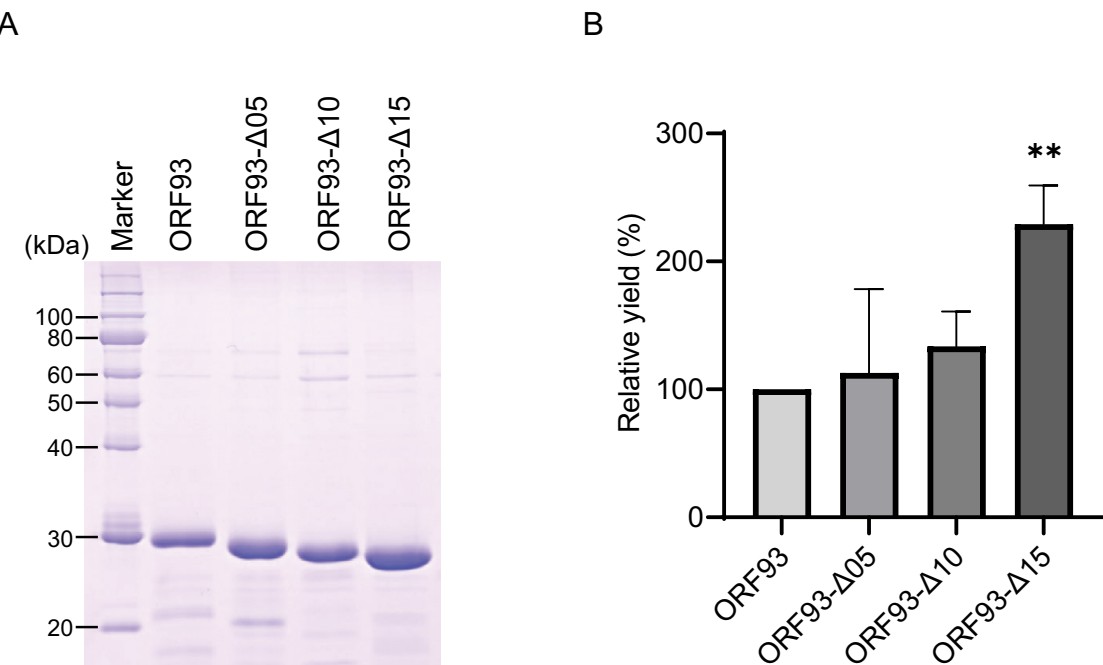

**Fig 2. Production of ORF93 and its mutants.** (A) SDS-PAGE image. We used the protein marker (XL-Ladder Broad protein molecular weight, APRO Life Science Institute) as a standard, of which thin bands at 20, 40, 50, 60, and 100 kDa and thick bands at 30 and 80 kDa contain 0.15 μg and 0.6 μg proteins, respectively. After elution from the affinity chromatography and filtration, 10 μL of elutes were electrophoresed. (B) Yields of ORF93 and its mutants. We evaluated a yield as a relative rate, of which the ORF93 is set as 100%. Error bars indicate SD (n = 4). The yield of ORF93 was statistically compared with those of linker deletion mutants. A double asterisk shows statistical significance (**, $P < 0.01$).

**Table 1. Examination of transparency of spot formed by ORF93 and its mutants against various bacteria using a double-layered agar plate assay.**

| Tested bacteria | | ORF93 | ORF93-Δ05 | ORF93-Δ10 | ORF93-Δ15 |
|---|---|---|---|---|---|
| Species | Strains | | | | |
| S. aureus | SA27 | + | + | ++ | ++ |
| S. schleiferi | D91 | + | + | ++ | ++ |
| S. felis | C3 | + | + | ++ | ++ |
| S. sciuri | D108 | + | + | ++ | ++ |
| S. arlettae | D119 | + | + | ++ | ++ |
| S. pettenkoferi | C58 | + | + | + | ++ |
| S. kloosii | D187 | + | + | + | + |
| S. pseudintermedius | D19 | ± | ± | + | + |
| S. epidermidis | GTC289 | ± | ± | ± | + |
| B. subtilis | 168 | - | - | - | - |
| E. coli | JM109 | - | - | - | - |

The degree of transparency of the spot was examined at a 4-level scale. The levels of transparency on the spot are ranked at "strong," "moderate," and "weak" as "++", "+", and "±", respectively; no transparency of spot is shown as "-".

the three mutants to the tested bacteria, all the tested proteins showed the transparency of spots against all nine strains of *Staphylococcus* spp.; these proteins did not form a transparency of spot on *B. subtilis* and *E. coli* strains (Table 1). Thus, the antimicrobial spectra of ORF93 and the three mutants were not different from each other. Of note, ORF93-Δ10 and ORF93-Δ15 showed stronger transparency of spots than ORF93 and ORF93-Δ05, suggesting that they have high antimicrobial activity.

Next, we measured the time-course bacterial turbidity change, and calculated and compared the bacteriolytic activities. In this study, the bacteriolytic activity was put as the reduction in turbidity per unit time and dose ($\Delta OD_{595}$/min/mg; S5 Fig). Four bacteria, including *S. aureus*, *S. epidermidis*, *S. pseudintermedius*, and *S. schleiferi*, were tested. As a result, similar to the result in the double-layered agar plate assay described above, bacteriolytic activities tended to increase as the linker was shortened (Fig 3). The ORF93-Δ15 had significantly higher bacteriolytic activities than other endolysins against the tested four *Staphylococcus* spp. Compared with ORF93, ORF93-Δ15 had ca 1.9, 2.1, 2.9, and 1.9 times higher bacteriolytic activities against *S. aureus*, *S. epidermidis*, *S. pseudintermedius*, and *S. schleiferi*, respectively ($P < 0.0001$). Thus, although the linker adjustment of ORF93 did not confer the antimicrobial spectrum, it was likely to improve the bacteriolytic activity against *Staphylococcus* spp.

## Long-term stability in solution at cold temperature

Storing ORF93 and three mutants at 4°C, the protein quantities over time were measured for 56 days. In this study, we measured the protein quantity of protein solution, after centrifugation to remove the insoluble proteins, and how the relative rate of protein quantity, which at the first day was set as 100%, was calculated. The protein quantities of the tested endolysins overall decreased by ca. 45–65% on 56 days (Fig 4A), because proteins in the storage tended to turn to insoluble forms. On day 14, the average relative protein quantities of ORF93, ORF93-Δ05, ORF93-Δ10, and ORF93-Δ15 were ca. 76%, 69%, 74%, and 85%, respectively; on day 56, those were ca. 55%, 48%, 35%, and 52%, respectively. We then compared these protein quantities on days 14 and 56 (Fig 4B). On day 14, the protein quantities did not differ among ORF93 and three mutants; on day 56, the protein quantity of ORF93-Δ10 was significantly lower than that of ORF93 ($P < 0.01$), while those of the other two mutants were not. In addition, by analyzing the proteins on day 56 by SDS-PAGE, we confirmed the endolysin band; the thickness

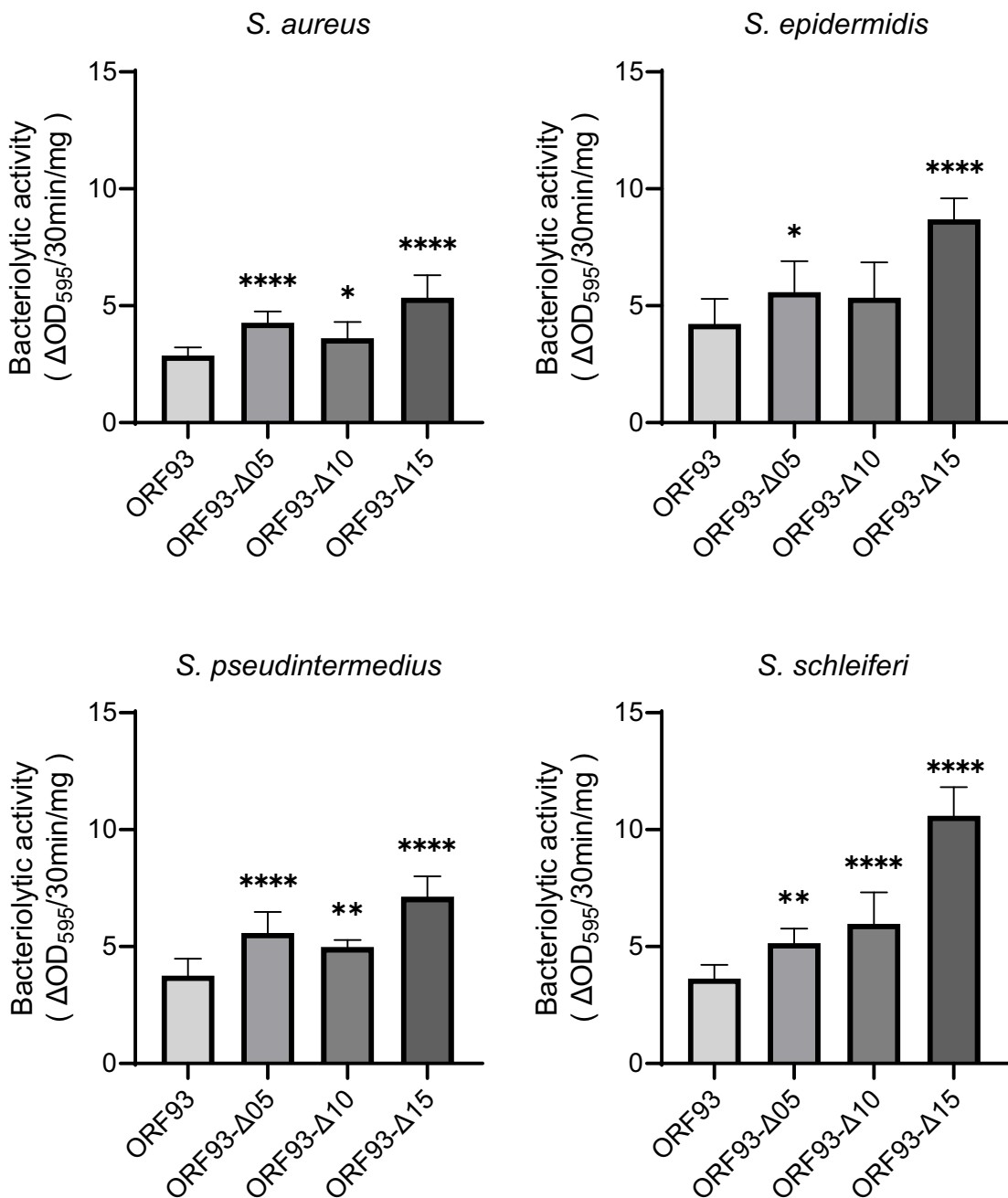

**Fig 3. Bacteriolytic activities of ORF93 and its mutants against *Staphylococcus* spp.** Error bars indicate SD (n = 12). We compared the bacteriolytic activities of ORF93 with those of linker deletion mutants. Statistical significance is shown on the graph as asterisks (****, $P < 0.0001$; **, $P < 0.01$; *, $P < 0.05$).

of minor bands slightly decreased as the linker length was shortened (Fig 4C). Thus, the linker is also important in ORF93 stability.

## Discussion

Efforts to develop endolysin as an antimicrobial agent in the pharmaceutical industry are already under way, and clinical trials are in progress [33,34]. In the endolysin development,

A

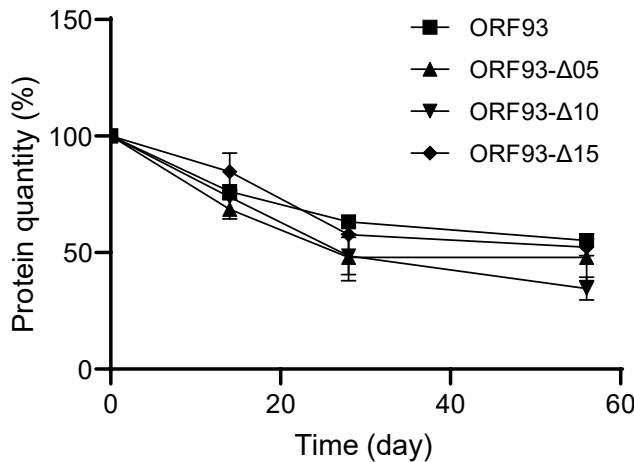

B

C

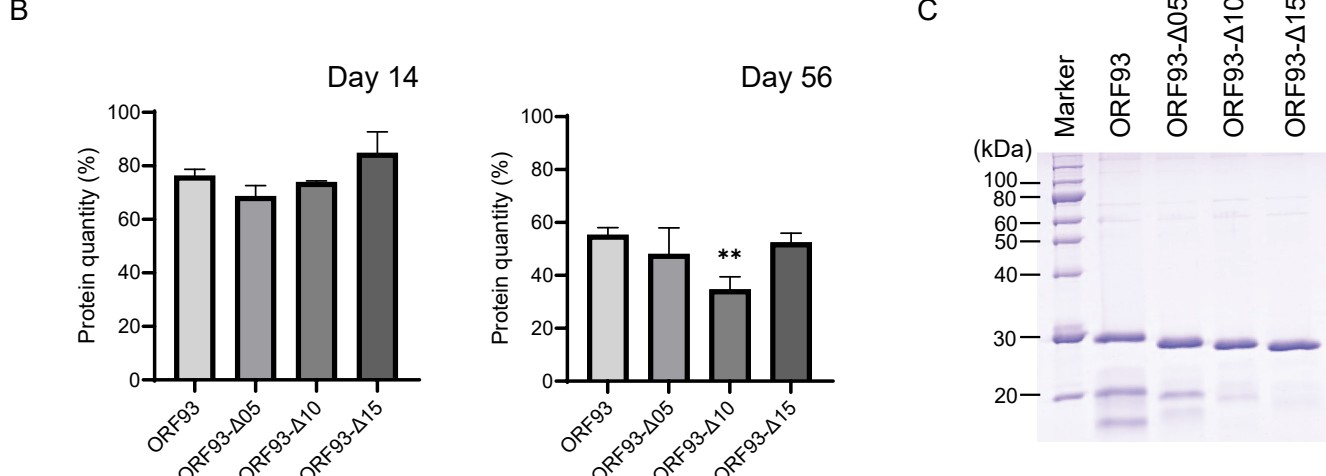

**Fig 4. Cold storage of ORF93 and its mutants.** Error bars indicate SD (n = 3). (A) Protein quantity changes over time. The protein quantity is presented as a relative rate, which at the first day was set to be 100%. (B) The protein quantity of the endolysin solution on day 14 and day 56. We compared protein quantity of ORF93 with those of linker deletion mutants on days 14 and 56. Statistical significance is shown on the graph with a double asterisk (**, $P < 0.01$). (C) SDS-PAGE image. We analyzed 10 μL of endolysin solutions by SDS-PAGE together with a protein marker (5 μL of XL-Ladder Broad protein molecular weight, APRO Life Science Institute) on day 56.

natural endolysin against *S. aureus* is fused and/or replaced with a protein domain from other bacteriolytic enzyme, and improves the antimicrobial activity and their yields [14–16,35]. To generate endolysins with improved activity and yield is difficult so that they have to be screened using a high-throughput system [36]. To extend the endolysin development, it is necessary to search for simpler and more effective modifications.

ORF93 is a novel endolysin derived from *Staphylococcus* giant phage S6, and is expected to be a potential antimicrobial agent against various pathogenic staphylococci. ORF93 is a two-domain endolysin, in which CHAP and SH3 domains are connected with a long atypical

linker, considering that multidomain proteins have an average linker length of 10 amino acids [37]. In addition, the ORF93 linker is atypically lysine-rich. In this study, we designed and produced the mutant endolysins by deleting the linker length of ORF93. Although the mutants such as ORF93-Δ05, ORF93-Δ10, and ORF93-Δ15 were produced, the mutant ORF93-Δ20 was not, which has the shortest linker of the endolysin tested in this study. One possible reason for the unsuccessful production of ORF93-Δ20 is that excessive deletion of the linker region led to the deletion of a rigid region close to the domain and resulted in inappropriate folding. The space provided by the linker is also important for folding during the protein translation process [38,39]. The protein with an extensively short linker may fail protein folding, because the domains come into contact with each other due to linker length during folding. Thus, the linker length to some extent is required to be optimally adjusted in the endolysin mutant production. We evaluated the influence of linker length adjustment on endolysin characteristics, such as yield, bacteriolytic activity, and storability using ORF93 and its three mutants.

The yield can be important to be considered in the endolysin development, because large production capability in *E. coli* can lead to the cost performance improvement in manufacturing. In this study, the relative yields tended to increase as the linker was shortened, and the ORF93-Δ15 exhibited the highest yield and purity. As *E. coli* has several proteases, the recombinant proteins may be proteolyzed by these enzymes [40]. The proteases recognize and cleave specific protein sequences presented on the protein surface. The shorter the linker sequence, the less frequent the recombinant protein is proteolyzed [41,42]. In particular, the linkers of ORF93 and its mutants are located outside of the protein, and changing the linker length could have affected the susceptibility to proteolysis. Thus, the reduction in linker length can contribute to improving the yield.

The high bacteriolytic activity and spectrum are also important in endolysin development to achieve high treatment efficiency against various bacterial infections. In this study, ORF93-Δ15 exhibited the highest bacteriolytic activity among the tested endolysins. One possible reason for this effect can be that the length and flexibility of the linker can change the geometric orientation between domains [35]. On the other hand, the antimicrobial spectrum was not influenced by the linker shortening. A previous study has shown the production of endolysin with bacteriolytic activity against both *Staphylococcus* spp. and *Streptococcus* spp., by replacing the peptidoglycan-binding domain of the streptococcal endolysin LambdaSa2 with that of staphylococcal endolysin [43]. If the antimicrobial spectrum of endolysin is required to be modified, the domain shuffling can be a solution.

The cold chain is crucial for commercial distribution [44]. Particularly, the prevention of natural aggregation is important when protein is stored [45]. In this study, testing the long storability at the cold temperature, ORF93 and its mutants were not natively stable. In addition, the minor protein bands of ORF93 increased as the protein was stored for a long period. The minor protein bands decreased, as the linker length was shortened. The degradation by carry-over molecules from the *E. coli* expression was assumed to occur on ORF93 and its mutants with a longer linker; ORF93-Δ15 did not seem to be influenced. In general, both domain and linker are important in protein stability. First, it is reasonable to consider that the protein domain functions in protein stability. The study has indicated that protein stability is related to domain molecular characteristics, such as hydrogen bond, hydrophobic interactions, and disulfide bond [46]. Moreover, the linker is also important in protein stability. Previous studies have demonstrated that the linker modification contributes to the thermal and pH stability of proteins and may impact their overall stability [24,47,48]. To enhance the storability of endolysin, delicate improvements to both the linker and domain may be required.

In general, endolysin has been optimized to a phage life cycle through viral evolution, whereas its function as an enzyme has not been maximized. Phage evolves the cell lysis, such

as the timing of lysis and the lysis system optimal to the progeny phage release. The expression of endolysin starts in the middle of the phage infection process [9]. If a phage expresses endolysins with high bacteriolytic activity in the middle phase of the infection process, progeny phages cannot be produced efficiently. In addition, phage S6 is a giant phage, and a longer linker of ORF93 may function to produce larger holes in the peptidoglycan by simply extending the distance from CHAP to SH3 domain.

To understand the role of the linker further, it is also important to examine both linker extension and modification. A previous study showed that a longer linker improves the thermostability of subunit-fused nitrile hydratase; however, an excessively long linker can decrease its activity and expression [47], which is consistent with the results of this study. Moreover, in protein engineering, various linkers have been designed and classified as flexible, rigid, helical, and cleavable linkers [19,49–51]. It has been reported that changing the type of linker can change the stability and activity of the enzyme [38,47,48]. In the future, the linker types will be examined in ORF93-Δ15.

To study the feasibility of endolysins, they must be investigated from other aspects, such as physical characteristics and therapeutic effects in vivo. First, physical characteristics such as pH stability and thermostability are important to develop an endolysin drug, because protein instability leads to irreversible aggregation and is a major concern in the development of new biopharmaceuticals. In particular, these characteristics have been investigated in detail to develop monoclonal antibodies as biopharmaceuticals [52,53]. Similarly to this, the physical characteristics of endolysins must be investigated regarding their suitability in biopharmaceuticals in the future. In addition, the additives enabling the protein stability are also important, to extend the usability to an endolysin drug. For example, a previous study has shown that the activity of LysK endolysin was maintained at 100% when stored at 22˚C for one month with polyols such as glycerol, whereas without polyols, it decreased to 5% [54]. Moreover, the therapeutic effects in vivo may be varied depending on endolysin type. In the previous study, some endolysins have shown therapeutic effects equivalent to vancomycin in mice with systemic MRSA infections, whereas other endolysins have not [12]. In addition, it has been shown that endolysins exhibit different levels of biofilm removal activity [12]. We need to investigate the therapeutic effects in vivo with an understanding of the biochemical properties of endolysins.

In conclusion, the optimization of the linker length of ORF93 improved the yield, and bacteriolytic activity against *Staphylococcus* spp., which suggested that it could be improved from wild-type ORF93 as an antimicrobial agent. This study addresses the subject more comprehensively than previous articles and shows that optimizing linker length could have multiple benefits for endolysin development. We propose that the linker investigation can be one of the key processes in endolysin development.

## Materials and methods

### Prediction of protein structure

The domain structure was predicted using the InterPro software package (https://www.ebi.ac.uk/interpro/) [29]. Protein conformation was predicted using AlphaFold2 within ColabFold [31,55]. The protein conformation was visualized using UCSF ChimeraX (https://www.rbvi.ucsf.edu/chimerax) [56].

### Bacterial strains, culture media, culture conditions, and reagents

*E. coli* JM109 (Champion® 109 High, SMOBIO Technology, Inc., Hsinchu City, Taiwan) was used for cloning. TaKaRa Competent Cells *E. coli* BL21 (Takara Bio, Shiga, Japan) was used for protein expression.

To examine the antimicrobial activity and bacteriolytic activity, the following bacteria were used: *S. aureus* SA27, *S. schleiferi* D91, *S. pettenkoferi* C58, *S. felis* C3, *S. arlettae* D119, *S. pseudintermedius* D19, *S. epidermidis* GTC289, *S. kloosii* D187, and *S. sciuri* D108 [27]; *B. subtilis* 168 [28]; *E. coli* JM109 (Champion® 109 High, SMOBIO Technology, Inc.).

All strains were cultured in Luria-Bertani-Miller broth (LB broth, Kanto Kagaku, Tokyo, Japan) at 37˚C. Ampicillin 100 μg/ mL was supplemented to the LB broth if needed.

All reagents used in this study were purchased from FUJIFILM Wako Pure Chemical Industries, Ltd (Osaka, Japan) unless otherwise stated.

## Construction of expression vector

The protein sequence of ORF93 was downloaded from GenBank (accession no. BDE75631). The DNA sequence optimized to *E. coli* was designed using Invitrogen GeneArt gene optimizer [57]. Synthetic genes were outsourced to Thermo Fisher Scientific (Waltham, MA, USA; S6 Fig). The synthetic gene was subcloned into pUC19 cloning vector, using the In-Fusion® HD Cloning Kit (Takara Bio). Inverse PCR was performed with primer sets (S1 Table) using a pUC19 with the synthetic gene as a DNA template, to produce the mutant synthetic genes. The PCR products were then connected, using the In-Fusion® HD Cloning Kit. The genes with accurate sequence were digested with *Sac* I and *Xba* I, and cloned into the pCold II vector (Takara Bio), of which the protein fused with translation enhancing element and 6×His tag at the N-terminus can be expressed.

## Protein expression and purification

Protein expression was performed using pCold II vector (Takara Bio), according to the company's instructions. Briefly, LB broth supplemented with ampicillin, containing *E. coli* BL21, which carries the gene sequence of the endolysin, was incubated at 37˚C with a shake until 0.4 to 0.5 of the optical density at 600 nm ($OD_{600}$). After cooling down to 15˚C and supplementing isopropyl-$\beta$-D-thiogalactopyranoside (IPTG) at 1 mM, the culture was incubated at 15˚C with a shake for 24 h. The cultures were centrifuged ($8,000 \times g$, 4˚C, 5 min), the cell pellets were resuspended in a sonication buffer (100 mM sodium phosphate, 300 mM NaCl, pH 7.8), and sonicated (12.7 mm diameter horn tip, 16% amplitude, 5-s pulses at 5-s intervals 30 times) on the ice, using a Q700 sonicator (Qsonica, Newtown, CT, USA). The cell lysates were centrifuged ($9,000 \times g$, 4˚C, 20 min), and the supernatants were incubated with $Co^{2+}$ agarose resin (ProteNova, Kagawa, Japan) for 2 h at 4˚C. After washing the resin with sonication buffer, the proteins were eluted with 2 mL of elution buffer (100 mM sodium phosphate, 300 mM NaCl, 300 mM imidazole, pH 7.8). The eluates were filtered through a 0.22 μm membrane filter (Millex® GV; Millipore, Billerica, MA, USA).

## SDS-PAGE and protein quantification

The protein solution was mixed with a 3× Laemmli sample buffer (6% w/v SDS, 30% v/v glycerol, 0.03% w/v bromophenol blue, 0.3 mol/L dithiothreitol, 180 mmol/L Tris-HCl, pH 6.8), and were heated at 95˚C for 5 min. The samples were analyzed in a SDS-PAGE gel (12.5% w/v acrylamide) together with protein molecular weight markers (XL-Ladder Broad; APRO life Science Institute, Tokushima, Japan). The gel was stained with Coomassie Brilliant Blue R250 to confirm protein purification.

The theoretical molecular mass was calculated from the protein sequence using GENE-TYX® Ver. 14.1.1 (Nihon server, Tokyo, Japan); the protein molecular mass was estimated from the thickest band in the SDS-PAGE image using ImageJ software (version 1.54g) [58].

Protein purification was examined by measuring the thicknesses of the bands in the SDS-PAGE images using ImageJ software (version 1.54g); the protein quantity in the SDS-PAGE image from 10 kDa to 250 kDa was set to 100%; and the proportion of protein band was calculated.

After the centrifugation (12,100 × g, 15 seconds) of protein solution, the supernatant was subjected to protein quantification using the TaKaRa Bradford Protein Assay Kit (Takara Bio).

## Measurement of antimicrobial activity using a double-layered agar plate assay

A hundred microliters of the overnight bacterial liquid culture were mixed with 5 mL of LB broth soft agar (LB broth supplemented with 0.5% w/v agar) on a 1.5% w/v LB agar plate. After the soft agar had solidified, 3 μL of endolysin solution (0.1 mg/mL) or elution buffer as a negative control was dropped onto the surface of the double-layered agar plate. After incubation at 37˚C for 12 to 24 h, the degree of the transparency of spots was visually examined at a 4-level scale (strong, moderate, weak, and no activity; S4 Fig).

## Measurement of bacteriolytic activity

Bacteria were grown in LB broth until the mid-log phase. The bacterial solutions were then washed three times with Dulbecco's phosphate-buffered saline (pH 7.4) and adjusted to an $OD_{600}$ of 1.0. The washed bacterial solutions were dispensed at 190 μL per well into 96-well microplates, together with 10 μL of endolysin solution (0.05 mg/mL) or elution buffer as a negative control. Optical density at 595 nm ($OD_{595}$) was measured immediately with a shake at 37˚C over time, using a Multiskan FC® (Thermo Fisher Scientific).

## Measurement of protein quantities during cold storage

Endolysin solutions were stored at 4˚C, and the protein quantities were quantified over time. SDS-PAGE and protein quantification were performed as described above.

## Statistical analysis

One-way ANOVA was used for the analysis, followed by Tukey's test or Dunnet test for the post hoc test. $P < 0.05$ was considered to indicate a significant difference. GraphPad Prism 4.0 (GraphPad Software Inc., San Diego, CA, USA) was used for the analysis.

## Supporting information

**S1 Fig. Protein sequences of ORF93 and its mutants.** The linker protein sequences are highlighted in yellow. (A) ORF93. (B) ORF93-Δ05. (C) ORF93-Δ10. (D) ORF93-Δ15. (E) ORF93-Δ20.
(PDF)

**S2 Fig. Predicted protein structure of ORF93 mapped with pLDDT score.** The colors on the protein structure indicate pLDDT score: < 50 in red, 50–90 by gradient color from orange, yellow, green to light blue, respectively, and > 90 in blue. The region with higher pLDDT scores indicates the structure with a higher confidence model.
(PDF)

**S3 Fig. Predicted aligned error (PAE) plot for ORF93.** In the PAE, the scored residue *x* with regards to the aligned residue *y* is shown as a heatmap. The color scale shows the expected

position error, for which the unit is Å.
(PDF)

**S4 Fig. Examination of antimicrobial activity on a double-layered agar plate.** (A) Assessment criteria for the degree of the transparency of spot. The transparency of spot was evaluated at a 4-level scale: ++, strong; +, moderate; ±, weak; and -, no activity. (B) Example of antimicrobial activity examination. *S. pettencoferi* C58 was tested.
(PDF)

**S5 Fig. Formula and schematic diagram for bacteriolytic activity.** In this study, we calculated the bacteriolytic activity from the above equation using the $OD_{595}$ value at 30 min.
(PDF)

**S6 Fig. *orf93* gene sequence used in this study.** The linker DNA sequence is highlighted in yellow.
(PDF)

**S1 Table. List of the PCR primers used in this study.**
(DOCX)

**S1 Raw images. Raw data of SDS-PAGE gel stained with Coomassie Brilliant Blue R250.** The gel images were taken with an EPSON GT-9800F scanner (Seiko Epson Corporation, Nagano, Japan). Original gel images of Figs 2A (A) and 4C (B).
(PDF)

## Acknowledgments

Molecular graphics and analyses performed with UCSF Chimera X, developed by the Resource for Biocomputing, Visualization, and Informatics at the University of California, San Francisco.

## Author Contributions

**Conceptualization:** Jumpei Uchiyama.

**Formal analysis:** Sosuke Munetomo, Iyo Takemura-Uchiyama.

**Funding acquisition:** Jumpei Uchiyama.

**Investigation:** Sosuke Munetomo, Iyo Takemura-Uchiyama, Hideyuki Kanda.

**Methodology:** Jumpei Uchiyama.

**Project administration:** Jumpei Uchiyama.

**Resources:** Jumpei Uchiyama.

**Supervision:** Jumpei Uchiyama.

**Validation:** Sosuke Munetomo.

**Visualization:** Sosuke Munetomo.

**Writing – original draft:** Sosuke Munetomo.

**Writing – review & editing:** Jumpei Uchiyama, Iyo Takemura-Uchiyama, Thamonwan Wanganuttara, Yumiko Yamamoto, Toshihiro Tsukui, Hideharu Hagiya, Shuji Kanamaru, Hideyuki Kanda, Osamu Matsushita.

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
