## [Decision Letter · Decision Letter 0]

23 Jul 2024

PONE-D-24-17026Examination of yield, bacteriolytic activity and cold storage of linker deletion mutants based on endolysin S6_ORF93 derived from *Staphylococcus* giant bacteriophage S6PLOS ONE

Dear Dr. Uchiyama,

Thank you for submitting your manuscript to PLOS ONE. After careful consideration, we feel that it has merit but does not fully meet PLOS ONE’s publication criteria as it currently stands. Therefore, we invite you to submit a revised version of the manuscript that addresses the points raised during the review process.

Both reviewers consider your manuscript as a valuable contribution to the field of endolysin research. However, there are minor changes to be done for the authors before the manuscript can be published. The academic author apologizes for the delay of publication procedure, in part also due to the difficulty to find suitable reviewers of the field.

We look forward to receiving your revised manuscript.

Kind regards,

Kornelius Zeth, PhD

Academic Editor

PLOS ONE

Journal Requirements:

2. Thank you for submitting the above manuscript to PLOS ONE. During our internal evaluation of the manuscript, we found significant text overlap between your submission and previous work in the [introduction, conclusion, etc.].

Please revise the manuscript to rephrase the duplicated text, cite your sources, and provide details as to how the current manuscript advances on previous work. Please note that further consideration is dependent on the submission of a manuscript that addresses these concerns about the overlap in text with published work.

[If the overlap is with the authors’ own works: Moreover, upon submission, authors must confirm that the manuscript, or any related manuscript, is not currently under consideration or accepted elsewhere. If related work has been submitted to PLOS ONE or elsewhere, authors must include a copy with the submitted article. Reviewers will be asked to comment on the overlap between related submissions (http://journals.plos.org/plosone/s/submission-guidelines#loc-related-manuscripts).]

We will carefully review your manuscript upon resubmission and further consideration of the manuscript is dependent on the text overlap being addressed in full. Please ensure that your revision is thorough as failure to address the concerns to our satisfaction may result in your submission not being considered further.

"This work was funded by a joint research fund of Nippon Zenyaku Kogyo Co., Ltd."

"This work was funded by a joint research fund of Nippon Zenyaku Kogyo Co., Ltd. Molecular graphics and analyses performed with UCSF Chimera X, developed by the Resource for Biocomputing, Visualization, and Informatics at the University of California, San Francisco, with support from National Institutes of Health R01-GM129325 and the Office of Cyber Infrastructure and Computational Biology, National Institute of Allergy and Infectious Diseases."

"This work was funded by a joint research fund of Nippon Zenyaku Kogyo Co., Ltd."

"No authors have competing interests."

7. We are unable to open your Supporting Information file [S1-S6_Fig]. Please kindly revise as necessary and re-upload.

Reviewers' comments:

Reviewer's Responses to Questions

**Comments to the Author**

1. Is the manuscript technically sound, and do the data support the conclusions?

Reviewer #1: Yes

Reviewer #2: Yes

2. Has the statistical analysis been performed appropriately and rigorously? 

Reviewer #1: Yes

Reviewer #2: Yes

3. Have the authors made all data underlying the findings in their manuscript fully available?

Reviewer #1: Yes

Reviewer #2: Yes

4. Is the manuscript presented in an intelligible fashion and written in standard English?

Reviewer #1: Yes

Reviewer #2: Yes

5. Review Comments to the Author

Reviewer #1: Munetomo et al., characterized three linker deletion mutants ORF93-delta05, ORF93-delta10, and ORF93-delta15 of endolysin S6_ORF93 derived from Staphylococcus giant bacteriophage S6.

I agree with Authors that optimization of the linker sequence is necessary to obtain a desirable chimeric protein. There are a lot of studies showing that the optimal linker sequence length is crucial for high protein activity (see for example https://doi.org/10.1007/s00253-008-1468-4, https://doi.org/10.1016/j.molimm.2003.08.006, https://doi.org/10.3390/pr8121587 or other). Therefore, I support publication of the manuscript in the PLOS ONE Journal after minor revision.

Specific comments:

‘Amino acid residues’ sounds strange, 'amino acids' or 'residues', please correct throughout the manuscript, line 95, 99, 100 etc.

The authors state that ‘the molecular masses of ORF93, ORF93-delta05, ORF93-delta10, and ORF93-delta15 were estimated to be 28.6 kDa, 27.9 kDa, 27.4 kDa, and 26.9 kDa, respectively’ or ‘the theoretical molecular masses were 31.3 kDa, 30.7 kDa, 30.2 kDa, and 29.7 kDa’. What programs were used to determine the molecular masses of studied proteins?

What was the concentration of proteins used in bacteriolytic activity tests?, line 184

Reviewer #2: In the context of the prevalence of antibiotic resistance, it is crucial to develop alternative antibacterial strategies, and phage lysin is a good candidate molecule. However, there are many limiting factors in the development of phage lysins at present, and more basic research is needed. This work examined the influence of linker modification on the characteristics of the endolysisn derived from Staphylococcus jumbo phage S6, and found that ORF93-△15 showed the highest yield and bacteriolytic activity with a slight effect on the cold storage performance.

Several concerns:

1. Figure 1 and Figure 2 need to be merged into one image because they illustrate the same problem, so there is no need to separate them into two separate images.

2. How long can the modified protein maintain activity at room temperature (e.g. 23 degrees)?

3. It is recommended to conduct further experiments on the modified content elements, such as temperature stability, pH stability tests, biofilm removal tests, in vivo antibacterial tests, etc.

4. In addition, how does extending the linker, altering or optimizing the linker's amino acid sequence affect lysin?

6. PLOS authors have the option to publish the peer review history of their article (what does this mean?). If published, this will include your full peer review and any attached files.

Reviewer #1: No

Reviewer #2: **Yes: **Shuguang Lu

---

## [Author Response · Author response to Decision Letter 0]

2 Sep 2024

Responses to Reviewer #1’s comments

We have responded to one general comment and three specific ones and revised the manuscript accordingly.

Response to the general comment

1. Reviewer #1 recommended including some references supporting the fact that the optimal linker is crucial to the protein function. We have revised the manuscript to strengthen our manuscript as follows, with the addition of the suggested references.

Line 82: “The influence of linker … been examined well, to our knowledge.” has been replaced with the following sentences.

“For example, the expression of protein without a linker was not successful in Escherichia coli [20]. Changing the codons of the linker improved the expression of fusion protein [21]. The linker length is shown to be an important factor for protein characteristics such as stability [22], activity [23], and catalytic efficiency [24]. However, the influence of linker length on the endolysin characteristics has not been examined well, to our knowledge.”

Line 88: “Escherichia coli” has been replaced with “E. coli.”

Response to specific comments

2. Reviewer #1 suggested that “amino acid residues” be replaced with “amino acids.” We agree with this suggestion and have made this change throughout the text.

Lines 94, 95, 99-100, 106, 111, 115, 122, and 138: “amino acid residue” has been replaced with “amino acid”.

Lines 35, 100-101, 109, 111, 122, and 264: “amino acid residues” has been replaced with “amino acids”.

3. Reviewer #1 requested us to mention the software used to determine the molecular masses of studied proteins. We have modified the manuscript as follows.

Line 383: The following sentence has been added.

“The theoretical molecular mass was calculated from the protein sequence using GENETYX® Ver. 14.1.1 (Nihon server, Tokyo, Japan); the protein molecular mass was estimated from the thickest band in the SDS-PAGE image using ImageJ software (version 1.54g) [58].”

4. Reviewer #1 requested us to mention the protein quantity used in bacteriolytic activity tests in line 184. We have modified the manuscript as follows.

Line 184: “In the assay, the endolysin … examined after incubation.” has been replaced with the following sentences.

“In the assay, the endolysin solutions (3 µL of 0.1 mg/mL) were applied on the surface of double-layered culture media containing bacteria, and the transparency of the spot formed on the agar plate was examined after incubation.” 

Responses to Reviewer #2’s comments

Reviewer #2 made 4 comments. We have responded to these comments and revised the manuscript accordingly.

1. Reviewer #2 suggested merging Figs 1 and 2. We have done so, and this is now included as a new Fig 1. We have modified the manuscript as follows.

Line 114: “Fig 1. Description of ORF93. (A) Schematic … correspond to Fig 1A.” has been replaced with the following sentences.

“Fig 1. Description of ORF93 and its mutants. The domains and linker are colored as follows: CHAP domain in green, linker in red, and SH3 domain in yellow. (A) Schematic diagram of ORF93 based on the primary structure. Each number represents the amino acid position from the N-terminus at both ends of the domain, counted from the N-terminus. (B) Predicted protein structure of ORF93. (C) Schematic diagram of ORF93 mutants based on the primary structure. The linker deletions are shown as shadowed areas. The numbers around shadowed areas represent the amino acid position from the N-terminus at both ends of the deletion area. ORF93-Δ05, ORF93-Δ10, ORF93-Δ15, and ORF93-Δ20 are the ORF93 mutants which 5, 10, 15, and 20 amino acids were deleted in the upstream direction from the 157th amino acid from the N-terminus. The predicted linker lengths are 21, 16, 11, and 6 amino acids, respectively. (D) Predicted protein structures of ORF93-Δ05, ORF93-Δ10 and ORF93-Δ15. These protein structures were overlapped based on the CHAP domain. (E) Predicted protein structure of ORF93-Δ20.”

Line 121: “Fig 2A” has been replaced with “Fig 1C”.

Line 127: “Fig 2B” has been replaced with “Fig 1D”.

Line 129: “Fig 2C” has been replaced with “Fig 1E”.

Line 132: “Fig 2. Description of ORF93 mutants. The domains and linker … protein structure of ORF93-Δ20.” has been deleted.

Line 151: “Fig 3A” has been replaced with “Fig 2A”. 

Line 160: “Fig 3” has been replaced with “Fig 2”. 

Line 175: “Fig 3B” has been replaced with “Fig 2B”. 

Line 214: “Fig 4” has been replaced with “Fig 3”.

Line 222: “Figure 4” has been replaced with “Fig 3”.

Line 232: “Fig 5A” has been replaced with “Fig 4A”.

Line 236: “Fig 5B” has been replaced with “Fig 4B”.

Line 241: “Fig 5C” has been replaced with “Fig 4C”.

Line 243: “Fig 5” has been replaced with “Fig 4”

2. Reviewer #2 addressed the stability of the proteins at room temperature (i.e., how long can the modified protein maintain activity at room temperature?). We appreciate this comment. The storability can extend the usability of endolysins. We consider this information to be highly appropriate when discussing general endolysin characteristics. Thus, we have modified the manuscript as follows.

Line 322: A new paragraph has been added as follows.

“To study the feasibility of endolysins, they must be investigated from other aspects, such as physical characteristics and therapeutic effects in vivo. First, physical characteristics such as pH stability and thermostability are important to develop an endolysin drug, because protein instability leads to irreversible aggregation and is a major concern in the development of new biopharmaceuticals. In particular, these characteristics have been investigated in detail to develop monoclonal antibodies as biopharmaceuticals [52, 53]. Similarly to this, the physical characteristics of endolysins must be investigated regarding their suitability in biopharmaceuticals in the future. In addition, the additives enabling the protein stability are also important, to extend the usability to an endolysin drug. For example, a previous study has shown that the activity of LysK endolysin was maintained at 100% when stored at 22 °C for one month with polyols such as glycerol, whereas without polyols, it decreased to 5% [54]. Moreover, the therapeutic effects in vivo may be varied depending on endolysin type. In the previous study, some endolysins have shown therapeutic effects equivalent to vancomycin in mice with systemic MRSA infections, whereas other endolysins have not [12]. In addition, it has been shown that endolysins exhibit different levels of biofilm removal activity [12]. We need to investigate the therapeutic effects in vivo with an understanding of the biochemical properties of endolysins.”

3. Reviewer #2 recommended conducting further experiments, such as temperature stability, pH stability tests, biofilm removal tests, in vivo antibacterial tests, etc. We appreciate these comments. However, our intention is to conduct these experiments in the future, because of the limits to the manuscript length and thus its content. Instead, we have reviewed the work on endolysins again, and have added these points to the new paragraph described right above in the response to comment No. 2 of Reviewer #2.

4. Reviewer #2 raised questions on the effects of extending the linker, and of altering or optimizing the linker’s amino acid sequence on endolysin function. We have modified the manuscript as follows.

Lines 317–321: “Moreover, phage S6 … to maximize endolysin function [14-16, 32, 33, 40, 47]” has been replaced with “In addition, phage S6 is a giant phage, and a longer linker of ORF93 may produce larger holes in the peptidoglycan by simply extending the distance from CHAP to the SH3 domain.”

Line 321: A new paragraph has been added as follows.

“To understand the role of the linker further, it is also important to examine both linker extension and modification. A previous study showed that a longer linker improves the thermostability of subunit-fused nitrile hydratase; however, an excessively long linker can decrease its activity and expression [47], which is consistent with the results of this study. Moreover, in protein engineering, various linkers have been designed and classified as flexible, rigid, helical, and cleavable linkers [19, 49-51]. It has been reported that changing the type of linker can change the stability and activity of the enzyme [38, 47, 48]. In the future, the linker types will be examined in ORF93-Δ15.”

Minor modifications

Through this revision, we have found minor mistakes in the abstract. Thus, we have modified our manuscript as follows.

Line 32: "linker modification" has been replaced with "linker shortening".

Line 36: "all the endolysins" has been replaced with "ORF93 and its mutants".

Lines 658–659: “The vertical and horizontal … of ORF93, respectively” has been replaced with “In the PAE, the scored residue x with regards to the aligned residue y is shown as a heatmap”.

---

## [Editor Report · Decision Letter 1]

11 Sep 2024

Examination of yield, bacteriolytic activity and cold storage of linker deletion mutants based on endolysin S6_ORF93 derived from *Staphylococcus* giant bacteriophage S6

PONE-D-24-17026R1

Dear Dr. Uchiyama,

We’re pleased to inform you that your manuscript has been judged scientifically suitable for publication and will be formally accepted for publication once it meets all outstanding technical requirements.

Kind regards,

Kornelius Zeth, PhD

Academic Editor

PLOS ONE
---

## [Editor Report · Acceptance letter]

11 Oct 2024

PONE-D-24-17026R1 

PLOS ONE

Dear Dr. Uchiyama, 

I'm pleased to inform you that your manuscript has been deemed suitable for publication in PLOS ONE. Congratulations! Your manuscript is now being handed over to our production team.

Kind regards, 

on behalf of

Dr. Kornelius Zeth 

Academic Editor

PLOS ONE